# Mass spectrometry and Monte Carlo method mapping of nanoparticle ligand shell morphology

Zhi Luo [1], Yanfei Zhao [1], Tamim Darwish[2], Yue Wang[1], Jing Hou[1] & Francesco Stellacci [1,3]

Janus, patchy, stripe-like, or random arrangements of molecules within the ligand shell of nanoparticles affect many properties. Among all existing ligand shell morphology characterization methods, the one based on mass spectroscopy is arguably the simplest. Its greatest limitation is that the results are qualitative. Here, we use a tailor-made Monte Carlo type program that fits the whole MALDI spectrum and generates a 3D model of the ligand shell. Quantitative description of the ligand shell in terms of nearest neighbor distribution and characteristic length scale can be readily extracted by the model, and are compared with the results of other characterization methods. A parameter related to the intermolecular interaction is extracted when this method is combined with NMR. This approach could become the routine method to characterize the ligand shell morphology of many nanoparticles and we provide an open access program to facilitate its use.

[1] Institute of Materials, École Polytechnique Fédérale de Lausanne, 1015 Lausanne, Switzerland. [2] The National Deuteration Facility, Australian Nuclear Science and Technology Organisation, 2232 Kirrawee DC, NSW, Australia. [3] Interfaculty Bioengineering Institute, École Polytechnique Fédérale de Lausanne, 1015 Lausanne, Switzerland. Correspondence and requests for materials should be addressed to F.S. (email: francesco.stellacci@epfl.ch)

The ligand shell imparts functionalities to nanoparticles (NPs) because it allows various different molecules to form the particle's outer layer[1–4]. It has been shown that additional properties derive from the close packing[5,6] of such molecules and their supramolecular organization[7–11]. The latter we call ligand shell morphology (LSM) hereafter. As an example, we can discuss NPs surface hydrophobicity. It can be tuned by either adopting ligands with varying polarity or using a mixture of ligands[12–14]. In the latter case results vary significantly depending on the arrangement of the ligands[15,16]. For binary mixtures, Janus, patchy and stripe-like morphologies would provide very different hydrophobic profiles to the NPs. In order to make use of the predicted large morphological diversity[17–20], characterization techniques that are easily accessible and versatile are needed. So far, several techniques, such as scanning tunneling microscope (STM), small angle neutron scattering (SANS), nuclear magnetic resonance (NMR), matrix assisted laser desorption/ionization (MALDI), and electron paramagnetic resonance (EPR), have been explored to characterize the structure of mixed self-assembled monolayers (SAMs) on NPs, i.e., the LSM[21–25]. Most of these methods can only provide qualitative information about the morphology. Spectroscopic techniques need a full series of NPs with varying composition to reach a qualitative conclusion[26–28]. The only two methods that can quantitatively characterize the ligand shell structures are STM[29–32] and SANS[33]. While STM can be used to extract length scales of stripe-like domains, it falls short in describing any other morphology. Moreover, due to the requirement of high-resolution images, only a few NPs are measured, resulting in uncertainties over the whole population. Currently, SANS is the only method that offers a comprehensive determination of any morphology over the whole sample. It takes advantage of the Monte Carlo based methods to fit the scattering curves and generate 3D models that can describe the organization of the ligand shell[34,35]. However, there are several stringent requirements for this technique, e.g., NPs with high monodispersity and very high solubility are required. Both STM and SANS are not suitable for routine and rapid characterization. STM requires careful sample preparation as well as time consuming optimization of imaging conditions[29,31], while the availability of synchrotron beamtime and the need for deuterated ligand molecules hinders the routine application of SANS.

Among all the existing methods, MALDI-TOF MS holds great promises due to its wide accessibility as well as simplicity in the measurements[36–41]. The application of MS to probe the structure of LSM takes advantage of the fragmentation phenomenon of NPs in MALDI-TOF[40,42]. It has been reported that certain fragments composed of metal–ligand complexes are prone to be desorbed from the NPs surfaces during the matrix assisted ionization process[43]. A general chemical formula of the fragment can be represented as $M_kL_n$, where M is the metal atom and L is the ligand molecule and $k$ and $n$ are the numbers of metal atoms and ligand molecules in the fragment, respectively. In the case of NPs coated by binary mixtures of ligand, a series of the fragment species can be detected in the mass spectra, i.e., the $M_kL_xL'_{n-x}$ fragments with the index $x$ being an integer number from 0 to $n$. It has been shown that by studying all the fragments at constant $n$ (and consequently $k$) it is possible to gather structural information on the LSM[36,37]. Indeed, the distribution of the fragments with $x$ from 0 to $n$ follows a binomial distribution when the two ligands are randomly distributed within the ligand shell. The more patchy-type segregation the LSM presents, the more the fragmentation profile differs from the binomial distribution, with the Janus morphology being the limiting case. The method has been reported to characterize mixed SAM structures on both gold and silver NPs and was utilized to follow the mechanism of ligand exchange reaction on NPs[44]. Current data analysis of MALDI-

TOF MS uses a single number, i.e., the sum-square-residue (SSR), to evaluate the statistical distance of the fragment distribution from the random morphology. When compared to binomial distribution, some threshold values for SSR are then arbitrarily assigned to classify the LSM into random, patchy, and Janus[36]. Such summarization of the whole MALDI spectra into one value is an over-simplification that here we show how to overcome.

Inspired by the data analysis methods used to reconstruct 3D NPs models by fitting SANS data[34,45], here we have developed a Monte Carlo type fitting approach to produce 3D LSM models by fitting all the fragmentation data present in a MALDI-TOF spectrum, i.e., not limiting ourselves at a single $n$. The method offers a rapid way to characterize different types of LSM. We show that a 3D model generated by fitting the MALDI-TOF data agrees well with that extracted from the fitting of the SANS data. Furthermore, we demonstrate that this fitting approach provides detailed information on the LSM. For example, we have used the models to retrieve the nearest neighbor distribution for a series of NPs, which can be used to interpret the chemical shifts in NMR measurements of the nanoparticles.

## Results

**Monte Carlo analysis of MALDI-TOF MS.** The fragmentation of the nanoparticles in MALDI process can be regarded as a stochastic sampling of the SAM on nanoparticle surfaces, which forms the basis of the algorithm we propose. The distributions of different fragments contain information of both the ratio of the two types of ligands and the correlation between them (more precisely the nearest neighbor distribution). While the ligand ratio can be directly calculated by integrating the composition over all the fragments at any $n$[43], linking the fragment distribution with ligand organizations is not as straightforward. As shown in Fig. 1, here we have developed a fitting procedure that uses a Monte Carlo approach to compare the theoretical mass spectroscopy fragmentation profile of a NP with a given LSM to the experimental values. The method is designed to fit at the same time all the fragments that the experiments produce. This is a departure from current literature where the analysis is limited to a single $n$. Practically we use $n = 4$, 5, 6 for the calculation as these fragments could be reproducibly detected by MALDI-TOF MS from the silver and gold NPs with good resolution. The analysis has three key steps.

First, a spherical surface composed of $N$ number of uniformly distributed beads is generated. The size of the sphere and the number $N$ need to be determined by other techniques such as TEM and TGA. Each bead can be assigned either to the value 0 or 1, representing the two different ligands. Practically a random assignment of the values is used as the starting ligand organization while other types of specific morphologies could also be used and lead to the same final solution.

Then, the intensity distribution of different metal–ligand fragments is calculated from this starting model using a simulated fragmentation process of the NP surfaces. Taking silver NPs as an example, the general formula of the fragments can be expressed as $Ag_{n+1}L_xL'_{n-x}$, where $x$ is the number of L ligand in the fragment composed of $n$ ligands in total. The distributions of different types of fragments that are generated from each molecule and its nearest neighbors are first calculated. As established previously[38], one could assume that all the molecules on the nanoparticle surface have the same probability to be detached and form different fragments. Therefore, the fragmentation pattern from all the $N$ molecules are then summed to generate the probability distribution of each type of $Ag_nL_xL'_{n-x}$ fragment using Eq. (1):

$$\omega_{calc}^{n,x} = \frac{\sum_N \binom{n}{x} \cdot \varphi_L^x \cdot \varphi_{L'}^{n-x}}{N} \quad (1)$$

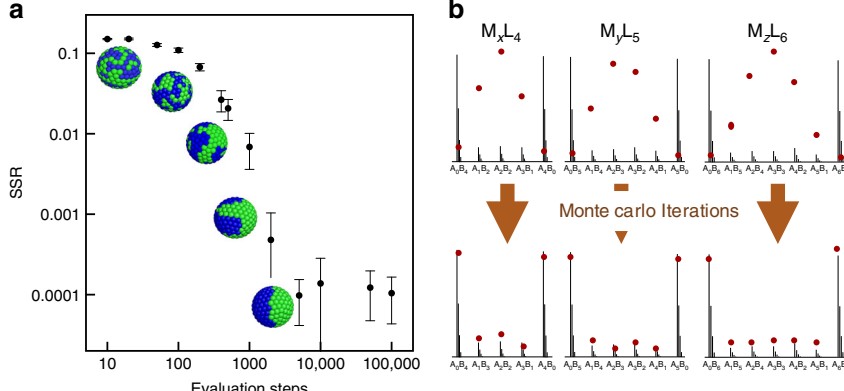

**Fig. 1** The Monte Carlo fitting of an idealized Janus type nanoparticle. **a** Efficiency and convergence of the Monte Carlo process. For every 1000 iteration steps, the computer calculation time required is around 2 min. Representative 3D models at each stage are shown. The error bars are the standard deviations of SSR values based on 10 replicated Monte Carlo runs. **b** Convergence of the fragmentation distribution patterns before and after the Monte Carlo process. MALDI spectra are simulated from perfect Janus type nanoparticle. Red dots indicate the fragmentation peaks of models from Monte Carlo calculation, starting from random configuration

where $\varphi_L$ and $\varphi_{L'}$ are the percentage of L and L′ molecules in the nearest neighbor of each molecule, represented by:

$$\varphi_L(k) = \frac{N_L(k)}{N_L(k) + N_{L'}(k)}, \ \varphi_{L'}(k) = \frac{N_{L'}(k)}{N_L(k) + N_{L'}(k)} \quad (2)$$

The discrepancy between the experimental mass distribution and that of the model generated is calculated and represented by the number, SSR:

$$SSR = \sum_{n=4}^{6} \frac{\sum_{x=0}^{n} (\omega_{calc}^{n,x} - \omega_{exp}^{n,x})^2}{n+1} \quad (3)$$

Equation 3 is then used as the scoring function for the final Monte Carlo process. The aim of the iteration is to minimize the SSR value. To start with, the value of a randomly selected bead in the model is flipped and the new fragment distribution patterns are calculated. If the SSR of the new model is lower than the previous one, then the change of value assignment is accepted. Otherwise the modification is rejected. The updated model is used again as the starting model for the next iteration step. Iteration could be stopped once the SSR value converges, which typically requires more than 100 N reconfigurations or the SSR value being below $10^{-4}$, as discussed below.

**Efficiency of the modeling**. The method was first tested using a simulated data from a model Janus nanoparticle with 4 nm core diameter and full coverage of ligands, i.e., total ligand number $N = 240$. As shown in Fig. 1, the three different fragmentation distributions for a perfect Janus nanoparticle are simulated and used as the input experimental data for the Monte Carlo program. The Monte Carlo program was run multiple times (>10) starting always from random configurations, i.e., binomial distribution and gradually converging to the input pattern. The error bars in Fig. 1a are the standard deviations of SSR values after certain iteration steps. After ~50 N (~10,000) iterations, the resulted model starts to show the same Janus feature as the original model. The final SSR is $9.9 \times 10^{-5}$, indicating the good quality of the fitting. The mass intensity distributions of different types of fragments are all very close to that of the input values as shown in Fig. 1b.

We show that the Monte Carlo fitting procedure (i.e., the convergence process) could also provide a better understanding

of the implications of SSR values. The program was run multiple times and the outcome at different number of iteration steps were recorded so that one could monitor the evolution of the morphologies together with SSR values during the fitting. Note that compared to previous literatures in which SSR values are calculated based on binomial distribution, here the SSR value represents the statistical distance between a model and a perfect Janus arrangement. As shown in Fig. 1a, the expected Janus feature was obtained after around 100 N iterations, which takes around 20-min computer time using for example MacBook with 2.9 GHz Intel Core i7. The SSR decreases rapidly, with ~1000 iterations already giving more than 2 magnitudes drop of the original SSR value, which takes less than 2 min of calculation time. The variations of the SSR at each stage are around 10%. As the final model shows practically the same features as the input Janus morphology, one can safely assume that the morphology of two types of nanoparticles are the same when the SSR is below or in the order of $10^{-4}$. When SSR value is around $10^{-3}$, the morphology looks close to large patchy type. Hence, similar to the threshold values of SSR against binomial distribution in the previous reports, one could conclude that when SSR values are above $10^{-3}$, detectable differences between the two morphologies could be seen. Significant deviations would present between two models when SSR values are above $10^{-2}$. Such assignment of SSR values is close to the threshold values used in previous literatures, but it is shown through an experiment for the first time here.

**Test of idealized geometry**. While Monte Carlo process often leads to trapped configurations at local minimum, the results for Janus morphology look rather robust. It might be due to the fact that the features and the corresponding mass spectra of Janus type nanoparticles are relatively unique. Therefore, we continued to test this algorithm using several other types of morphologies as the fitting target. As shown in Fig. 2, the Monte Carlo fitting could successfully retrieve different structural features of the ligand shell morphologies on nanoparticles. For patchy nanoparticles with several separated patchy domains or the one with two large domains, the reconstructed models show similar type of ligand organization as expected (Fig. 2a–c). The stripe-like morphologies could be modeled and captured as well (Fig. 2d) although the fitted model is not as perfect as the idealized stripes. Due to the ambiguity in the nature of the mass spectra, the relative positions of the patches as well as the symmetry of the

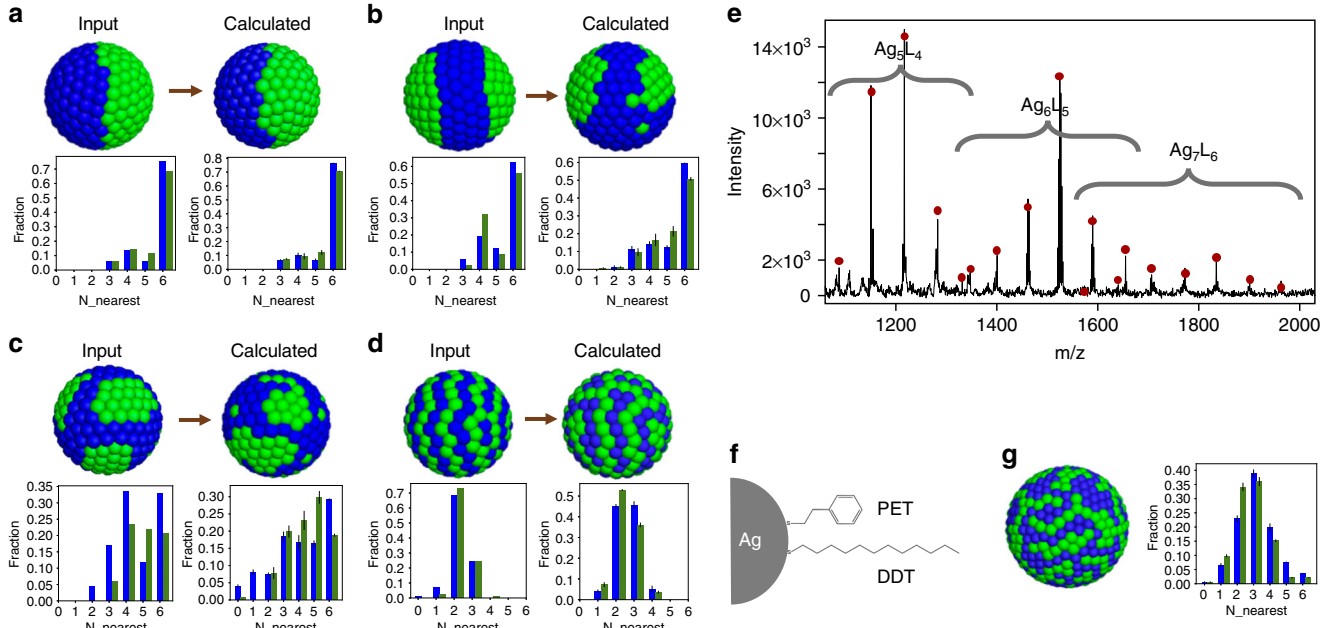

**Fig. 2** Test of the method on NPs with different LSMs. Input (left) and resulted model (right) of idealized LSMs featuring **a** Janus separation; **b** two large patchy domains; **c** separated patchy domains; **d** stripe-like domains are shown together with their nearest neighbor distribution profiles. The $x$ axes of the profiles stand for the number of nearest neighbors of ligand A with the same identity of A. The $y$ axes stand for the fractions of ligands with different nearest neighbors. The blue and green columns correspond to beads with blue and green colors, respectively. **e** MALDI-TOF MS data of silver NPs protected by PET and DDT ligands. Red dots mark the intensity for the fragmentation of calculated model. **f** Schematics of the chemical structure of the ligands. **g** Model and the corresponding nearest neighbor distribution from the Monte Carlo fitting of the MALDI spectra. Blue beads stand for DDT ligands while green beads stand for PET. The error bars are the standard deviations of nearest neighbor distribution based on five replicated calculations

patchy and stripe organizations are not exactly the same. Since the SSR values for the final fits are all below $2 \times 10^{-5}$, indicating a very good agreement in the fragmentation patterns, this result again points out the resolving power of the MALDI-TOF technique in discriminating mixed LSMs.

The 3D models only serve as a direct visualization of the structural similarities/differences between two LSMs. In fact, one can determine quantitative parameters of the LSM structures from the models. As discussed in the algorithm of mass spectra calculation, the fragmentation is indeed a stochastic sampling of the LSM and reflects the nearest neighbor distribution of the ligands. We thus developed an algorithm to retrieve such information. In Fig. 2a–d, the profiles of the first nearest neighbors with same identity of ligand A (green bead) are shown. While Janus LSM shows very high contribution of ligands surrounded by the same ligands ($x = 6$), the nearest neighbor distribution from stripe-like LSM features ligands that are at the interfaces ($x = 2–4$). For all the four cases, the resulted 3D model matches well with the input idealized configuration. Overall the data shown in Fig. 2 indicate that the method developed is very strong in retrieving an overall LSM, and in particular its strength is in determining the local structure within the LSM, i.e., the nearest neighbor distribution. The method does not have a true feedback for symmetry and in fact often the symmetry retrieved is lower than the input. Interestingly, the SANS method developed recently has the opposite characteristic with a signal that mostly based on the overall symmetry of the particles and much less on the local LSM composition[33]. Meanwhile, the fitting procedure is based on random switch of bead assignments and does not lead to higher symmetry. As an example, the program is used to fit the spectra coming from a random geometry. As shown in Supplementary Figure 1, the Monte Carlo calculation could

reproduce the random bead arrangement as well as the close to binomial distribution of nearest neighbors.

Furthermore, as one could notice from Fig. 2a–d that the models from Monte Carlo calculations do not always capture all the structural details in complex morphologies. This is unfortunately due to the intrinsic ambiguity in the structural information that the MALDI data contains. Different models could give very similar experimental footprint and thus the solution from Monte Carlo fitting is not unique. In order to address this limitation, the fitting program was repeated five times for each morphology. Each generated model can be regarded as a possible solution in the space of all possible structures corresponding to a given MALDI spectra. It is important to notice that the physical cause for the MALDI spectra is the nearest neighbor distribution, which is the most reliable result of our method. For all of the particles studied, we have generated standard deviations in nearest neighbor distributions. As shown in Fig. 2a–d, the standard deviations for all the calculations are small indicating that all the models from repeated calculations are very similar to each other. These standard deviations are the best way to estimate the resolution of our model. LSMs that produce nearest neighbor distribution that fall within these standard deviations will not be distinguishable with this approach. We should point out that our empirical observation is that SSR values that are within $10^{-4}$ to each other lead to indistinguishable nearest neighbor distributions.

**Fitting experimental data.** As the first experimental example, mixed ligand-protected silver NP was prepared using a modified Stucky method[46] at room temperature using dichloromethane as the solvent. The detailed synthesis and characterizations can be found in the Methods section. The NP has a core diameter of

$5.8 \pm 0.7$ nm, as measured by TEM (Supplementary Figure 2) based on counting of more than 500 NPs, are used as the first example. The nanoparticles were coated with phenylethanethiol (PET) and 1-dodecanethiol (DDT) (Fig. 2f). It can be estimated that there are ~500 ligands on the NPs surface, using a combination of TEM and TGA, Supplementary Figure 3. MALDI-TOF of the nanoparticle was measured in reflection positive mode and using trans-2-[3-(4-tert-Butylphenyl)-2-methyl-2-propenylidene] malononitrile (DCTB) as the matrix. As shown in Fig. 2e, fragment peaks corresponding to $Ag_5L_4$, $Ag_6L_5$, and $Ag_7L_6$, could all be resolved well. The mass accuracy was 50 ppm and the isotope patterns of each peak match perfectly with theoretical calculations as shown in an example in Supplementary Figure 4. The integrations of the peak intensities were performed and used as the input information for the fitting program. A detailed list of all the fragment masses as well as the regions used for the integration of each peak are listed in Supplementary Table 1. Before the Monte Carlo analysis, a consistency check on the ligand ratio was performed to make sure that there were no systematic errors among different fragments. Specifically, the ligand ratio $\theta$ could be calculated from each type of fragments from the Eq. (4) below:

$$\theta = \sum_{i=0}^{n} i * P_i \text{ with } P_i = \frac{n_i}{\sum_{i=0}^{n} n_i} \qquad (4)$$

where $n_i$ is the intensity of fragments containing $i$ ligand L; $P_i$ is the fraction of the fragment containing $i$ ligands L; $\theta$ is the ratio of the ligand L. If the integration of all the intensities is accurate, the extracted ligand ratio from each type of fragment should be practically the same. Indeed, as shown in Supplementary Figure 5 and Supplementary Table 2, the PET fraction varies from 54% to 56%, indicating the high consistency. In fact, previously Cliffel and co-workers[47] have compared the ligand ratios calculated from the fragmentation in MALDI-TOF MS with the NMR measurements for nanoparticles with varying ligand shell hydrophobicity and bulkiness. An average difference of less than 1% in relative abundance was reported in their work. Similarly, as shown in Supplementary Table 3, for the nanoparticles we used in this paper, the ligand ratios calculated from the two techniques match very well with <5% differences. These results indicate that fragments with the same $n$ (metal atoms) but different ligand combinations ionize with the same efficiency during the MALDI process.

The main difference of fitting of experimental data compared to idealized models is the presence of errors in the measured of peak intensity and integration. Therefore, in the Monte Carlo calculation, instead of simply using SSR as the scoring function, the normalized SSR value by the errors of each peak in the spectra should be used as the new scoring function (SF)[34].

$$SF = \sum_{n=4}^{6} \frac{\sum_{x=0}^{n} ((\omega_{calc}^{n,x} - \omega_{exp}^{n,x})/\sigma_{integ})^2}{n+1} \qquad (5)$$

The errors of the mass peak integration are calculated using the RMSD (root-mean-square deviation) of the baseline noise[48]. Specifically:

$$\sigma_{integ}^2 = \sigma_i^2 * N \qquad (6)$$

where the $\sigma_{integ}$ is the standard deviation of the integration of peak intensity, $\sigma_i$ is the standard deviation of the baseline noise, $N$ is number of points in the region of integration.

Monte Carlo calculation was then applied to the experimental data using this new scoring function and $10^4$ iteration steps was performed yielding a final SSR value of $5 \times 10^{-4}$. The slight

increase of final SSR values compared to the idealized geometry is due to the presence of errors in the experimental data. In the idealized morphology, the mass peak intensities of the outcome model overlap almost perfectly with the input spectra as there was no errors and uncertainties. The program was run for multiple times (>5) and all the resulted models look very similar while the standard deviation of nearest neighbor distribution is small.

As shown in Fig. 2g, the resulted model shows a complicated organization of the two ligands, featuring both small patches as well as stripe-like domains. As a result, the nearest neighbor distribution of this nanoparticle is close to that of idealized stripe-like LSM in Fig. 2d compared to other morphologies. This is due to the fact that no large patchy structures of the ligands are formed. To further understand the suitability of the nearest neighbor descriptor and its possible limitations in comparing different predictions, a more complex descriptor, i.e., nearest neighbor distribution in the first two neighboring shells (18 neighbors) of the two models are computed and compared. The same type of analysis was reported previously[49]. As shown in Supplementary Figure 6, in the 18 nearest neighbor distribution profile, the differences between the two structures become clearer. While the idealized stripe-like nanoparticle shows a more centralized distribution featuring the high fraction of 7–10 same nearest neighbors, the distribution profile for PET-DDT nanoparticle is broader.

**Quantitative comparison with SANS**. In order to further prove the validity of our method, we focus on the comparison between SANS and MALDI-TOF, as SANS is the only existing technique that can be used to quantitatively characterize mixed ligand shells with complicated morphologies. We chose to analyze with MALDI the same Ag NP that had been reported previously, i.e., Ag NPs protected with deuterated PET and DDT[33]. As shown in Fig. 3a, the previous SANS characterization indicates that the nanoparticle shows a patchy-type ligand shell distribution, with PET ligands forming patchy domains. MALDI-TOF analysis was also performed on the same NPs as reported before. The Ag NP has a core diameter of around 5.9 nm corresponding to a total number of $N = 510$ ligands. The resulted model from Monte Carlo calculation of the MALDI-TOF data is shown in Fig. 3b with the final SSR value being $2 \times 10^{-4}$. The two models show similar structural features, i.e., PET ligands forming patchy domains together with scattered distribution within DDT patches.

Both Monte Carlo analysis of SANS and MALDI data give low-resolution models, but the two techniques differ in the nature of the models that they generate. From SANS models, each bead only acts a space holder and represents the possibility of finding a

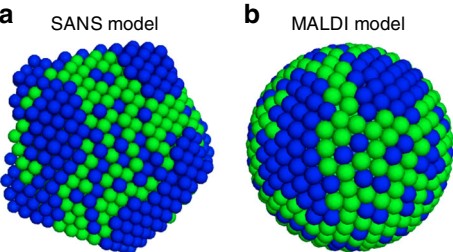

**a** SANS model  **b** MALDI model

**Fig. 3** Comparison between the SANS and MALDI models. **a** SANS model of the dPET and DDT protected silver nanoparticle that was reported previously. **b** 3D model from the Monte Carlo calculation of MALDI-TOF data. In both models, blue beads stand for PET ligands while green beads stand for DDT

certain type of ligand molecules. Therefore, SANS model represent an average molecular organization lacking single molecule details. On the other hand, models calculated from MALDI reflect the nearest neighbor distribution of the ligands. As one could not calculate nearest neighbor profile from SANS model, we used a method described previously to measure the lengths/width of the patchy domains of the two models[33]. The averaged width of PET domains is found to be $1.9 \pm 0.6$ nm and $1.7 \pm 0.6$ nm for the SANS and MALDI models, respectively. Such differences in value is again due to the different nature of the models. While in SANS one measures the distances on the top of the ligands (end functional groups), the measurements of MALDI model is based on the position of sulfur atoms and will be necessary smaller[3].

Furthermore, the matching between the SANS and the MALDI data also strongly suggests that the particles studied are monodisperse in LSM. In fact, it should be stated that both methods are affected by polydispersity in the LSM, but they are affected in different ways. The scattering pattern of a polydisperse sample is the average of all the form factors (shape), while MALDI spectra averages the nearest neighbor profiles. We show clearly this concept in Supplementary Figure 7, where the MALDI spectra of a mixture of different patchy models is fitted with our program. The resulted model presents the averaged patchy feature with interconnect patches. Yet, if one calculates the small angle neutron scattering (SANS) pattern of the particle mixture and the Monte Carlo model, significant differences could be seen (Supplementary Figure 7B) in the theoretical SANS pattern. Hence, while each method can produce a model for the particles (with MALDI being significantly easier), it is from their combination that one can get an answer on whether the sample is polydisperse or not. Were the sample to be polydisperse then a simultaneous ensemble fitting of both data could resolve such polydispersity. Fractionation methods could also help to separate different species in a sample[50] in order to be measured with MALDI separately.

**Interpretation of NMR chemical shifts**. As reported previously, the NMR chemical shift of ligands on nanoparticles depends on the ligand shell structure[26,28]. By plotting the trend of chemical shift against the ligand ratio, one can categorize the type of the LSMs to be random, stripe-like, and Janus types. Importantly both for the random (linear) and Janus ($1/x$) LSM, it was possible to find a formula to capture the trend using a single parameter. This was not possible for patchy particles as the dependence of the nearest neighbor distribution on the overall ligand composition was not known. Hence, in the previous paper, stripe-like particles were reported to have a sigmoidal type chemical shift trend, but it was not possible to determine whether a single parameter was sufficient to fit the data. Here, we use our Monte Carlo analysis of MALDI MS to quantify the nearest neighbor composition for 12 patchy NPs differing in composition, and show that indeed a single parameter is sufficient to describe all the NMR chemical shifts.

Assuming that the final chemical shift of the ligand on the nanoparticle is the sum over all the possible situations of its nearest neighbor distribution ($n_i$), one could get:

$$(\text{chemical shift})_L = \sum_{i=0}^{6} c_{L,i} * n_i \qquad (7)$$

where $n_i$ is the normalized probability of ligand L with $i$ number of L' ligands in its first nearest neighbor and $c_{L,i}$ is the corresponding chemical shift of ligand L. With an assumption

that the interaction potential between two ligands is additive, $c_{L,i}$ can be calculated with two parameters, i.e., $c_{L,0}$ being the chemical shift of homoligand L protected nanoparticle and $\sigma$ being the increment in chemical shift by having one L' in the vicinity of L. Therefore, the Eq. (7) becomes:

$$(\text{chemical shift})_L = \sum_{i=0}^{6} (c_{L,0} + i\sigma) * n_i \qquad (8)$$

As $\sum_{i=0}^{6} n_i = 1$, one can rewrite Eq. (8) as:

$$(\text{chemical shift})_L = c_{L,0} + \sigma \sum_{i=0}^{6} i * n_i \qquad (9)$$

The above formula reduces to a linear dependence on ligand composition for random distribution and to a function close to $1/x$ for Janus distribution as shown in Supplementary Figure 8. For a generic patchy particle one needs to know the LSM for every composition. We ventured to determine this composition for the series of silver NPs coated by PET and DDT. The nanoparticles were synthesized using the same procedure as described above therefore they have similar core sizes and only varies in the ratio of the two ligands. All the MALDI-TOF MS and NMR spectra are shown in Supplementary Figure 9. A table comparing the ligand ratio calculated from NMR and different fragments in MALDI-TOF spectra is presented as Supplementary Table 2, which shows high consistency of the calculated ligand ratios between the two techniques (<5% differences). Figure 4a shows all resulted Monte Carlo models of silver NPs with varying ligand ratio. The 3D models produced were then used to obtain the $n_i$ for all the particles. The $c_{L,0}$ could be measured directly from homoligand nanoparticle. Therefore, $\sigma$ was the only parameter left to interpret the chemical shift of the ligands.

The homoligand PET nanoparticle gave a chemical shift at 6.59 ppm, corresponding to the $c_{L,0}$ value. As shown in Fig. 4b, the [1]H NMR chemical shifts of aromatic hydrogens on PET shift downfield as the ligand ratio of PET decreases. The value $\sigma$ was then determined using Eq. (9). Remarkably, the calculated values of $\sigma$ for all the 12 nanoparticle samples are within 10% variation, i.e., $0.100 \pm 0.007$ ppm as listed in Fig. 4a. By using the averaged $\sigma$ value, the chemical shifts predicted by MALDI-TOF models agree well with the measured NMR data as demonstrated in Fig. 4c. Since $\sigma$ is mainly determined by the van der Waals forces between the two types of ligands[23], the combined analysis of NMR and MALDI-TOF further opens up a new way of quantifying the interactions between mixed ligands on nanoparticle surfaces.

In summary, we show here that by combining MALDI-TOF MS and Monte Carlo calculations, it is possible to quantitatively reconstruct 3D models of mixed SAM protected nanoparticles in a rapid and effective way. A user-friendly version of the fitting program will be released as an open access software. We have further tested the program for various other types of nanoparticles such as gold nanoparticles and silver nanoparticles protected with other types of ligands demonstrating the versatility of the method, as discussed in the Supplementary Discussion, Supplementary Figure 10–11 and Supplementary Table 4. Considering the easy accessibility of MALDI-TOF MS, the method reported here could transform the characterization of the ligand shell of NPs coated with mixed SAM from a formidable challenge into a routine measurement.

## Methods
**General**. All the chemicals were purchased from Sigma-Aldrich and used as received. Deuterated PET was provided by National Deuteration Facility of Australian Nuclear Science and Technology Organisation. [1]H NMR spectra were

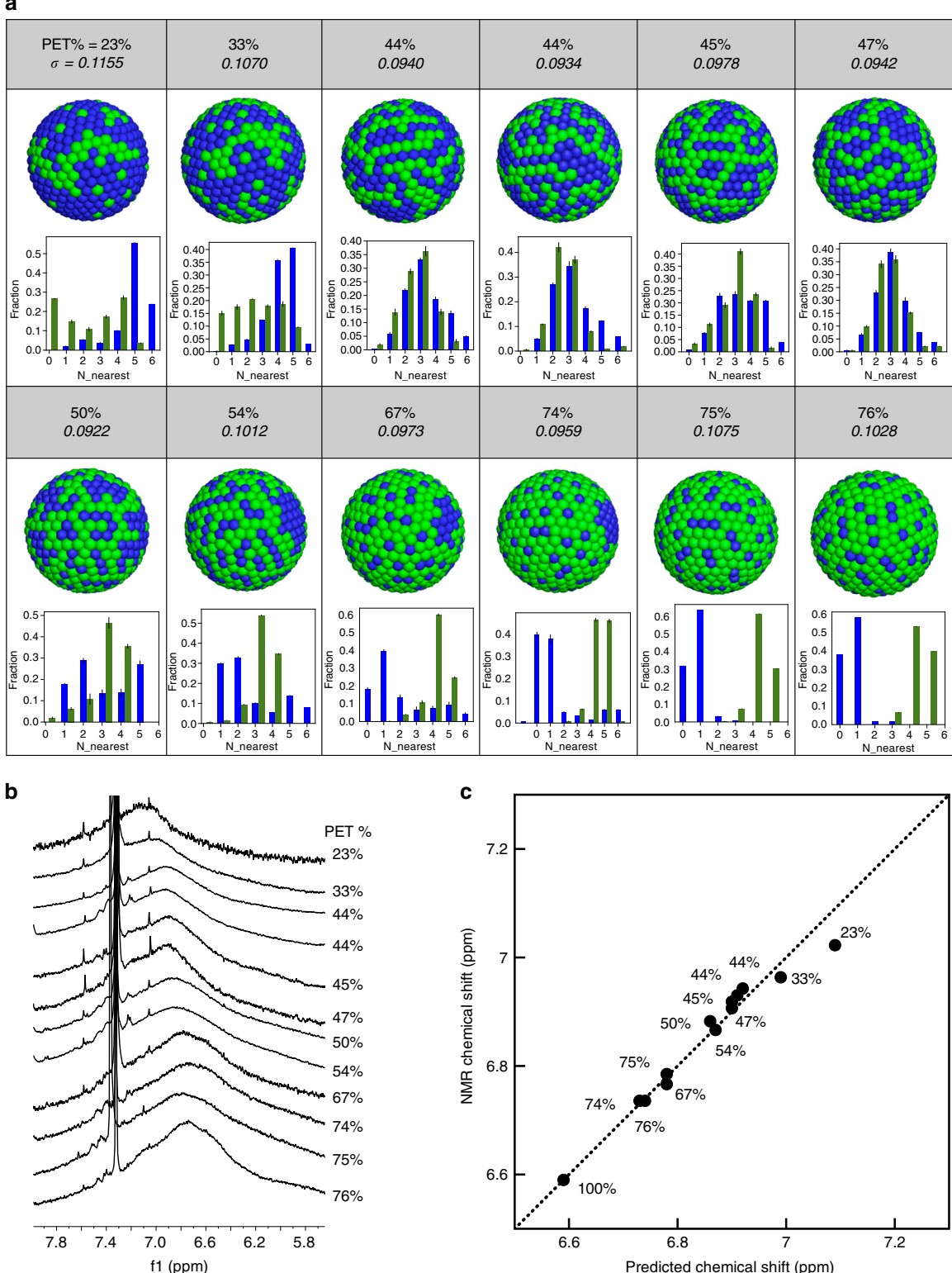

**Fig. 4** Comparison between NMR and MALDI-TOF results. **a** Models calculated from Monte Carlo fitting for PET-DDT silver NPs of varying ligand composition. Blue beads stand for DDT ligands while green beads stand for PET. **b** [1]H NMR for the aromatic hydrogens of PET Ligands. The x axes of the profiles stand for the number of nearest neighbors of ligand A with the same identity of A. The y axes stand for the fractions of ligands with different nearest neighbors. The blue column corresponds to beads with blue color. **c** Comparison between the chemical shifts calculated from nearest neighbor analysis of the MALDI model and that directly measured from [1]H NMR. Each data point corresponds to one PET-DDT silver NP sample, with the PET ligand ratio labeled. The error bars are the standard deviations of nearest neighbor distribution based on five replicated calculations

measured using Bruker 400 MHz instrument and deuterated dichloromethane was used as the solvent. TEM images were recorded with FEI Tecnai Osiris microscope. TGA measurements were performed using TGA 4000 instrument from Perkin Elmer.

**Synthesis of mixed SAM protected nanoparticles**. For the synthesis of silver nanoparticles, first 110 mg silver trifluoroacetate was added in 20 ml dichloromethane. A total of 0.5 mmol ligand molecules with varying composition were then added and the mixture was stirred for 10 min followed by adding 434 mg tert-butylamine-borane as the reducing agent and stir for 16 h at room temperature. Upon the completion of reaction, 40 ml methanol was added to quench the reaction. The nanoparticles precipitates were collected by centrifugation and were washed several times using methanol. The product was dried under vacuum overnight.

For the synthesis of gold nanoparticles, 50 mg gold(III) chloride trihydrate was fist dissolved in 40 ml ethanol followed by the addition of 0.25 mmol ligand mixtures. A total of 217 mg tert-butylamine-borane was then added and the reaction mixture was stirred overnight. The nanoparticles were precipitated and wash using diethyl ether several times and dried under vacuum.

**MALDI-TOF MS**. The MALDI measurements were performed using Bruker AutoFlex Speed instrument. DCTB was used as the matrix substance. To prepare for the sample, chloroform was used as the solvent to make a solution of the matrix with the concentration of 25 mg/ml. 1.0 mg nanoparticles sample was dissolved in 0.1 ml chloroform to make 10 mg/ml sample solution. Then the sample and matrix solution was mixed at volume ratio 1:1. 2 μl of the mixture solution was spotted on the stainless steel target plate. All the measurements were performed with positive ionization and reflection mode to detect in the 700–3500 mass range. The laser intensity was kept at 30% of the maximum. The resulted spectra were processed using FlexAnalysis software.

**Code availability**. An open access software that is used in the manuscript for the Monte Carlo calculations is available and can be downloaded from https://sunmil.epfl.ch. The source code is available from the authors upon request.

## Data availability

All raw data (MALDI-TOF MS in Figs. 2, 4, NMR of PET-DDT Ag NPs, TGA data and MALDI-TOF MS of gold NPs) are deposited and can be downloaded from the public data repository (Figshare.com) at [https://doi.org/10.6084/m9.figshare.7059236.v1].

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

## Acknowledgements

Z.L. and F.S. gratefully acknowledge funding from the Swiss National Foundation Division II. T.D. acknowledges the support from the National Collaborative Research Infrastructure Strategy—an initiative of the Australian Government. T.D. thanks Anwen Krause-Heuer for her help with the deuteration work. We thank Suiyang Liao for the help of TGA measurements.

## Author contributions

Z.L. and Y.Z. wrote the Monte Carlo program; Z.L., Y.W. and J.H. synthesized and characterized gold and silver nanoparticles; T.D. synthesized deuterated thiolated ligands; Z.L. and F.S. designed research and prepared the manuscript.

## Additional information

**Competing interests:** The authors declare no competing interests.

