## [Peer Review File · Nature Communications]

Reviewers' comments:

Reviewer #1 (Remarks to the Author):

The manuscript by Stellacci and co-workers aims to obtain precise maps of the distribution of ligands in the shell coating the surface of metal nanoparticles. Fragments obtained by MALDI analysis have a composition that reflects their distribution. As a consequence, as when solving a puzzle or sequencing DNA or proteins, it should be possible to piece together the whole map of ligands distribution.

The authors achieve this result by applying Monte Carlo simulations similar to those used to fit neutron scattering data. The distribution map obtained well compare to the results of other characterization methods and were used to predict the NMR chemical shift of a series of mixed shell nanoparticles.

The method proposed is extremely powerful and, as claimed by the authors, has the potential to become the routine method to characterize the ligand shell morphology. For this reason, the manuscript could be suitable for publication on Nature Communication.

There are, however, a few points that I'd like to see discussed in more detail.

The first regards the fragment generated. At first sight, it may appear surprising they precisely sample the surface local composition. Literature on fragments generation from shell-coated nanoparticles in MALDI analysis is apparently quite limited. Analysis of the references cited in the manuscript point to a series of papers from Cliffl and McLean where fragment formation was described and assumed to reflect surface composition. In subsequent papers, it is demonstrated that fragment composition give information on the shell morphology. In all these papers it is assumed, but never demonstrated, that all the fragments measured ionize with the same efficiency. It is however unclear which are the bases for such an assumption. Once formed, clusters as Au₄RS₄ and other are neutral and in order to be revealed they must become charged. Since the matrix used in most of the cases is DCBT, ionization occurs via one electron oxidation of the gold cluster. It is safe to assume that this electron transfer occurs with the same efficiency independently from the cluster composition? An experimental verification of this assumption could be based on the comparison of the monolayer composition obtained by MALDI fragment analysis with that obtained by other techniques. Even if already done in some of the early studies, such comparison should be performed and reported, as validation, for each sample studied. The authors compare the composition obtained by the analysis of different fragments in the first and last examples (supporting information) but not in the other ones, neither do their compare the MALDI compositions with those obtained with other techniques.

The second point regards the size dispersion of the nanoparticles. I agree with the authors that one of the advantages of this method is that nanoparticles with narrow size dispersion are not necessary since similar fragments with similar distribution will be produced by nanoparticles with different size. This however is true only if the morphology is size independent. If different morphologies are present in the sample depending on the particles size, this may likely affect the analysis. Of course, this argumentation evidences that another assumption is underlying the protocol described, i.e. that all the nanoparticles, even in the case of an extremely narrow size dispersion, have the same shell morphology (and composition) and no shell morphology (or composition) distributions are present.

This leads also to the third point that regards the resolution of the images obtained. The authors demonstrated in the first part that complex structures are not precisely reconstructed by the methods. This is not a major drawback since even in this case the mapping obtained is far better and more general than that of any other method so far reported. However it is possible that other morphologies

may give similar fragment distributions even if with slightly greater scorings. Was this taken into consideration? When we look to the model reported in figure 3, one can easily see that they look similar, but there are also a lot of subtle differences. Would they produce similar patterns? Which is the reliable piece of information that can be retained by the MALDI maps? On my opinion, the nearest neighbour plots are more solid than nanoparticle pictures.

Final point is also related. What is the effect of the errors on the integration of the mass peaks on the maps obtained? Inspection of figure 5 suggests that in the case of signals with low intensity or large clusters as Au5SR5 or Au6SR6 this may be huge. Still it is likely that composition distribution of the larger fragments may have a relevant effect on the morphology reconstructed, as they sample a larger surface portion.

Minor points:

1) In the supporting info characterizations are reported only for two samples, namely figures 2F and 5E. I presume that all the other samples analyzed (Figure 3, Figure 4, Figure 5A) are the very same of the original paper. In the case this is the case, this should be stated in the experimental section.

2) In the nearest neighbour plots of figure 4, report the bars for both the ligands, as in those of figures 2 and 5.

3) The statement of "thin stripe-like domains on the NPs surfaces" in the case of the nanoparticles reported in figure 5F is overstretched (only two stripes 4 ligand long are visible) and should be removed. Indeed, the nearest neighbour plot (Figure 5G, letter G is missing) indicates that most (80% at least) of the blue ligands form just monomers or dimers.

4) MALDI spectra for nanoparticles analyzed in figures 3, 4 and 5a should be reported in the ESI and it should be indicated which fragments were included in the analysis.

5) Distribution plots for the TEM pictures reported in the ESI should be reported (it looks unlikely that size dispersions of the particles in the two micrographs may be that similar, i.e. about 10% in both the cases, as in the second micrograph several very small particles are visible)

6) Fragment masses should be reported.

Reviewer #2 (Remarks to the Author):

The present manuscript by Stellacci and colleagues describes a methodology for using MALDI-TOFMS analyses of AuNP ligand distributions in conjunction with computer simulations to characterize the distribution of mixed-ligands on nanoparticle surfaces. The present work represents an extension of the series of works disseminated by Harkness et al. some years ago (Refs 26, 29, 31) along similar lines, but with several distinguishing differences. The present work is noteworthy in that it is demonstrated to also predict chemical shifts and correlations with SANS models are also presented. It also accounts for additional ligand species in the MS spectra, which may or may not improve the accuracy of the methodology. The addition of noise in the spectra and the impact on the conclusions drawn should be discussed. Along these lines, the present work makes no attempt to clean-up the "chemical noise" in the MALDI-TOFMS spectra, which was performed in the Harkness series using ion mobility as an ion pre-separation prior to TOFMS analyses. The Stellacci manuscript could be improved by inclusion of a paragraph comparing the previous works and strategies to those presently presented. This should definitely be included, but does not diminish enthusiasm for the present manuscript.

The extension of these strategies through monte carlo simulations for convergence to obtain a more objective characterization and merging these analyses with correspondence to NMR chemical shift prediction and SANS models are well conducted and provide a more universal approach to the general characterization of Janus or phase-segregated nanoparticle systems. This should be of keen interest to the readership of Nature Communications.

Reviewer #3 (Remarks to the Author):

The current manuscript represents a notable contribution to a very active field of monolayers self-assembled on nanoparticles, a field with numerous experimental and theoretical studies including some that are conflicting. The main idea of the manuscript is to rapidly determine surface arrangement/morphology of binary ligand mixtures through reverse Monte Carlo fitting of MALDI-TOF data. Using a simple geometric descriptor (essentially, nearest neighbor distribution of ligands), the authors successfully reconstruct surface morphologies that explain the experimental MALDI data and provide quantitative indicators of specific morphologies (e.g., patchy vs Janus). Additionally, the method is validated through comparison to SANS data and is shown to be able to derive empirical NMR relationships for mixed-ligand NPs. Leveraging MALDI-TOF combined with theoretical modeling for morphology determination is not novel in the field, but this manuscript presents a much simpler and more general approach to surface reconstruction and can be recommended to publication in Nature Communications. However, before it is published several omissions should be fixed.

Overall, plots of nearest neighbor distribution histograms should be revised or clarified. While I understand that the y-axis represents fractions of ligands, nowhere in the text or the plots themselves (e.g., y-label) is it mentioned. Similarly, I find the x-label ("Number") very vague. It is explained in the caption but I suggest using more scientific notation in the plots (e.g., N_{nearest}). Sticking to the histograms, can the authors comment on/show uncertainties for the data? Considering significant error bars shown in Fig. 1, it will be interesting to see how sensitive the nearest neighbor descriptor is to small variations in the surface arrangement. It should be easy to generate the error bars considering how fast MC fitting is. Similarly, it is unclear what the error bars in Fig. 1 are intended to represent. Do they come from multiple Monte Carlo runs? Are they block averages? This needs to be explained.

Comparing histograms and surface arrangement of the ligands in Figs. 1D and 1G, authors state that nearest neighbor distributions are very similar, which I agree with. However, I cannot fully agree that ligand morphologies are the same. The authors themselves say that particle G has considerable patches and stripes while calculated particle D exhibits only thin stripes. To further understand the suitability of the nearest neighbor descriptor and its possible limitations in comparing different predictions, the authors should look into more complex descriptors like nearest neighbor distribution in the first two neighboring shells instead of one (18 neighbors vs 6) like it was done in Fetisov and Siepmann, *J. Phys. Chem. B*, p. 1972, 2016. Although it will not change how the Monte Carlo fits are performed since only the first neighboring shell matters for MALDI-TOF spectra fitting, it can help to further disseminate the results or determine limitation of the method.

This leads me to the main omission in the presentation of the idealized Monte Carlo fitting results. It is well known that a main limitation of reverse Monte Carlo methods is the possibility of different models giving similar experimental footprint. While authors can successfully arrive to specific morphologies starting for random arrangements, what happens if one wants to fit to a random morphology? Is the developed MC method robust enough to reproduce the binominal distribution or will it possibly lead to

some higher symmetry?

On p.6 at the end of the second paragraph, I find the discussion about different SSR values a bit confusing and I suggest further clarification for readers. In the literature, SSR values are almost exclusively computed with respect to binominal distribution of ligands. In this manuscript, the authors discuss SSR with respect to idealized Janus topology and state that value of 10^{-3} would correspond to patchy NPs. However, this is only true if one uses Janus arrangement as the reference, but not binominal. This should be clarified since the SSR scale is flipped with respect to published trends.

When discussing quality of fits for different morphologies on p.7, the authors say that the SSR value for all fits was below 2×10^{-5} . Can authors explain why the fits are suddenly much better here, since throughout the whole text all SSR values were on the order of 10^{-4} or more, especially considering the accuracy (error bars)?

Overall, the section "Test on various samples" is contextually sparse. First, there is no mention of the experimental sizes and ligand ratios for the presented Ag NPs, which limits reproducibility. Secondly, the authors note that these NPs have been extensively studied in the literature but they do not discuss their findings in the context of the previous results. Are they consistent? Similarly, the paragraph about Au NPs does not bring anything new to the manuscript without comparing to previous findings. By this point, it is already clear that the approach developed by the authors can successfully reproduce experimental spectra and the mention of Au NPs detracts from the primary story. What would be more interesting is to discuss if, when applied to Au NPs, the new method also reveals something new or at least confirms previous findings/theories. Otherwise, I suggest to move this part to the Supporting Information. Also, the nearest neighbor distributions and the image of the Au NP suggest that the arrangement is more close to random distribution of monomers/"dimers" of DDT in MUA instead of stipe-like domains.

And some relatively minor but still important remarks:

- 1) On p.2, the authors write "ab initio calculations, i.e. Monte Carlo based methods". I believe there is a misuse of the phrase "ab initio" here that can lead to confusion.
- 2) On p.3, the authors discuss $M_k L_x L_n - x$ fragments and say that constant n leads to constant x . In the next sentence, they say that k varies from 0 to n to give a binominal distribution. Should not x and k be exchanged in this context?
- 3) On p.9, "to quantitatively characterization" should read "to quantitatively characterize".
- 4) Caption for Fig.5 is missing part (E) and instead has additional part (G).
- 5) On p.10, the first two sentences in the second paragraph have to be combined.

Response to reviewers' comments:

(The reply to the referee's questions/comments is in blue while the corresponding changes in the manuscript are in red.)

Reviewer #1 (Remarks to the Author):

The manuscript by Stellacci and co-workers aims to obtain precise maps of the distribution of ligands in the shell coating the surface of metal nanoparticles. Fragments obtained by MALDI analysis have a composition that reflects their distribution. As a consequence, as when solving a puzzle or sequencing DNA or proteins, it should be possible to piece together the whole map of ligands distribution.

The authors achieve this result by applying Monte Carlo simulations similar to those used to fit neutron scattering data. The distribution map obtained well compare to the results of other characterization methods and were used to predict the NMR chemical shift of a series of mixed shell nanoparticles.

The method proposed is extremely powerful and, as claimed by the authors, has the potential to become the routine method to characterize the ligand shell morphology. For this reason, the manuscript could be suitable for publication on Nature Communication.

We thank the referee for the highly positive assessments on the importance of our work.

There are, however, a few points that I'd like to see discussed in more detail.

The first regards the fragment generated. At first sight, it may appear surprising they precisely sample the surface local composition. Literature on fragments generation from shell-coated nanoparticles in MALDI analysis is apparently quite limited. Analysis of the references cited in the manuscript point to a series of papers from Cliffel and McLean where fragment formation was described and assumed to reflect surface composition. In subsequent papers, it is demonstrated that fragment composition gives information on the shell morphology. In all these papers it is assumed, but never demonstrated, that all the fragments measured ionize with the same efficiency. It is however unclear which are the bases for such an assumption. Once formed, clusters as Au₄RS₄ and other are neutral and in order to be revealed they must become charged. Since the matrix used in most of the cases is DCBT, ionization occurs via one electron oxidation of the gold cluster. It is safe to assume that this electron transfer occurs with the same efficiency independently from the cluster composition? An experimental verification of this assumption could be based on the comparison of the monolayer composition obtained by MALDI fragment analysis with that obtained by other techniques. Even if already done in some of the early studies, such comparison should be performed and reported, as validation, for each sample studied.

We agree with the referee that currently the mechanism of fragmentation and ionization of ligand shells on nanoparticle surfaces is not well understood. Previously, Cliffel and co-workers have reported the comparison of the ligand ratios obtained from NMR and mass spectrometry (*Anal. Chem.* 2010, 82, 9268–9274). They have analysed three types of nanoparticles with varying ligand shell hydrophobicity and bulkiness using DCTB as well as CHCA as the matrix. An average difference of less than 1% in relative abundance was reported in their work.

We should point out that NMR is currently the most efficient way to perform compositional analysis on the ligand shell of nanoparticles, we have reviewed the issue in a recent paper (*Acc. Chem. Res.*, 2017, 50, 8, 1911-9119).

Therefore, we have performed a series of NMR measurements on the nanoparticles with different ligand ratios used in this manuscript. A table comparing the ligand ratio from MALDI and NMR is shown in supporting information (Table S3). As shown in Table S3, the ligand ratio results from the two techniques agree well with less than 5% differences. Therefore, we believe that fragments with different ligand combinations indeed ionize with the same efficiency in the case of the particles shown in the paper. Were the fragments to ionize with different efficiency, one would need to correct for this difference in efficiency to obtain the right ligand shell distribution. We plan on adding this feature in future releases of our program. We will also upload all the raw MS and NMR data (currently shown in

Figure S9) to an open access data repository (figshare.com), as recommended by Nature Communications.

We have added the following paragraph in the main manuscript to clarify this issue:

In fact, previously Cliffler and co-workers²⁹ have compared the ligand ratios calculated from the fragmentation in MALDI-TOF MS with the NMR measurements for nanoparticles with varying ligand shell hydrophobicity and bulkiness. An average difference of less than 1% in relative abundance was reported in their work. Similarly, as shown in Table S3, for the nanoparticles we used in this paper, the ligand ratios calculated from the two techniques match very well with less than 5% differences. These results indicate that fragments with the same n (metal atoms) but different ligand combinations ionize with the same efficiency during the MALDI process.

The authors compare the composition obtained by the analysis of different fragments in the first and last examples (supporting information) but not in the other ones, neither do they compare the MALDI compositions with those obtained with other techniques.

In the same table (Table S3) above, we have presented the composition analysis from different fragments for all the samples. The ligand ratio obtained from different types fragments as well as NMR measurements agree well with each other (< 5% differences).

The following sentence was added to the manuscript:

All the MALDI-TOF MS and NMR spectra are shown in Figure S9. A table comparing the ligand ratio calculated from NMR and different fragments in MALDI-TOF spectra is presented as Table S3, which shows high consistency of the calculated ligand ratios between the two techniques (< 5% differences).

The second point regards the size dispersion of the nanoparticles. I agree with the authors that one of the advantages of this method is that nanoparticles with narrow size dispersion are not necessary since similar fragments with similar distribution will be produced by nanoparticles with different size. This however is true only if the morphology is size independent. If different morphologies are present in the sample depending on the particles size, this may likely affect the analysis. Of course, this argumentation evidences that another assumption is underlying the protocol described, i.e. that all the nanoparticles, even in the case of an extremely narrow size dispersion, have the same shell morphology (and composition) and no shell morphology (or composition) distributions are present.

The referee is correct, were the sample to be polydisperse in ligand shell morphology, then the result from the MALDI experiment would be an average of the morphologies present in the sample. We show clearly this concept in Figure S7, where the MALDI spectra of a mixture of different patchy models is fitted with our program. The resulted model presents the averaged patchy feature with interconnect patches. Yet, if one calculates the small angle neutron scattering (SANS) pattern of the particle mixture and the Monte Carlo model, significant differences could be seen (Figure S7B) in the theoretical SANS pattern. Therefore, this shows that the matching between the SANS and the MALDI data in Figure 3 strongly suggests that the particles studied are monodisperse in ligand shell morphology.

In this sense, one needs to appreciate that MALDI will be always a fast method for ligand shell identification, but the final answer has to be given by comparing with SANS plots (for monodisperse particles, or fractionated polydisperse ones). In fact, a possible second grade solution for polydisperse sample would be to produce different fractions of the nanoparticles via centrifugation with a sucrose gradient (Langmuir, 2015, 31 (41), pp 11179–11185) and have MALDI on the separate fractions.

We have added the following content to address this issue:

The matching between the SANS and the MALDI data also strongly suggests that the particles studied are monodisperse in ligand shell morphology. In fact, it should be stated that both methods are affected by polydispersity in the ligand shell morphology, but they are affected in different ways. The scattering pattern of a polydisperse sample is the average of all the form factors (shape), while MALDI spectra averages the nearest neighbor profiles. We show clearly this concept in Figure S7, where the MALDI spectra of a mixture of different patchy models is fitted with our program. The resulted model presents the averaged patchy feature with interconnect patches. Yet, if one calculates the small angle neutron scattering (SANS) pattern of the particle

mixture and the Monte Carlo model, significant differences could be seen (Figure S7B) in the theoretical SANS pattern. Hence, while each method can produce a model for the particles (with MALDI being significantly easier), it is from their combination that one can get an answer on whether the sample is polydisperse or not. Were the sample to be polydisperse then a simultaneous ensemble fitting of both data could resolve such polydispersity. Fractionation methods could also help to separate different species in a sample³⁷ in order to be measured with MALDI separately.

This leads also to the third point that regards the resolution of the images obtained. The authors demonstrated in the first part that complex structures are not precisely reconstructed by the methods. This is not a major drawback since even in this case the mapping obtained is far better and more general than that of any other method so far reported. However, it is possible that other morphologies may give similar fragments distributions even if with slightly greater scorings. Was this taken into consideration? When we look to the model reported in figure 3, one can easily see that the look is similar, but there are also a lot of subtle differences. Would they produce similar patterns? Which is the reliable piece of information that can be retained by the MALDI maps? On my opinion, the nearest neighbour plots are more solid than nanoparticle pictures.

We agree with the reviewer that the nearest neighbour distribution profile is a better and more quantitative description compared to the visual models. In fact, the MALDI spectra are directly related to the nearest neighbour profile. We have added text to strengthen this concept in the paper. Indeed, the issue of resolution should be addressed not looking at the models but analysing the nearest neighbour plots. In order to address this point, we have repeated the Monte Carlo calculations multiple times (> 5) for all the nanoparticles studied in this work. The generated models could be regarded as representations in the space of possible structures corresponding to each MALDI spectra. As shown in the updated Figure 2 and Figure 3, the nearest neighbour plots now have standard deviations calculated from these multiple runs. These standard deviations represent the resolution of our method.

We think that the resolution of MALDI models reflects its ability in distinguishing two similar structures. Different morphologies may give similar fragmentation patterns, which is the intrinsic ambiguity in the structural information that the MALDI data contains. In general, we observe that if SSR of fragmentation between two models differ approximately of 10^{-4} (or less), they result in very similar models. Therefore, an SSR threshold (10^{-4}) could be an effective evaluation of resolution of the method.

In order to clarify this issue, we have added the following:

Furthermore, as one could notice from Figure 1A-D that the models from Monte Carlo calculations do not always capture all the structural details in complex morphologies. This is unfortunately due to the intrinsic ambiguity in the structural information that the MALDI data contains. Different models could give very similar experimental footprint and thus the solution from Monte Carlo fitting is not unique. In order to address this limitation, the fitting program was repeated 5 times for each morphology. Each generated model can be regarded as a possible solution in the space of all possible structures corresponding to a given MALDI spectra. It is important to notice that the true physical cause for the MALDI spectra is the nearest neighbor distribution and this is the true result of our method. Hence for all of the particles studied, we have generated standard deviations in nearest neighbor distributions. As shown in Figure 1A-D, the standard deviations for all the calculations are small indicating that all the models from repeated calculations are very similar to each other. These standard deviations are the best way to estimate the resolution of our model. Ligand shell morphologies that produce nearest neighbour distribution that fall within these standard deviations will not be distinguishable with this approach. We should point out that our empirical observation is that SSR values that are within 10^{-4} to each other lead to indistinguishable nearest neighbor distributions.

Final point is also related. What is the effect of the errors on the integration of the mass peaks on the maps obtained? Inspection of figure 5 suggests that in the case of signals with low intensity or large clusters as Au5SR5 or Au6SR6 this may be huge. Still it is likely that composition distribution of the larger fragments may have a relevant effect on the morphology reconstructed, as they sample a larger surface portion.

We thank the referee for this comment. We now have improved the fitting algorithm taking into consideration the errors in the mass peak integration using an approach that is adopted by the Monte

Carlo fitting of SANS spectra (Biophysical journal 76, no. 6 (1999): 2879-2886). In the Monte Carlo calculation, instead of simply using SSR as the scoring function, we now use normalized the SSR with the errors of each peak in the spectra. The new scoring function (SF) for experimental data now becomes:

$$SF = \sum_{n=4}^6 \frac{\sum_{x=0}^n ((\omega_{calc}^{n,x} - \omega_{exp}^{n,x}) / \sigma_{integ})^2}{n + 1}$$

The errors of the mass peak integration are calculated using the RMSD (root-mean-square deviation) of the base line noise (Methods Mol Biol. 2010, 673, 211–222). Specifically:

$$\sigma_{integ}^2 = \sigma_i^2 * N$$

where the σ_{integ} is the standard deviation of the integration of peak intensity, σ_i is the standard deviation of the baseline noise, N is number of points in the region of integration.

The models and nearest neighbour distribution profiles in all the figures are now updated by the results from the new fitting algorithm. The features of the ligand shell morphologies as well as the nearest neighbour distributions obtained are very similar compared to the previous version.

We have changed/added the following content in the manuscript:

The main difference of fitting of experimental data compared to idealized models is the presence of errors in the measured of peak intensity and integration. Therefore, in the Monte Carlo calculation, instead of simply using SSR as the scoring function, the normalized SSR value by the errors of each peak in the spectra should be used as the new scoring function (SF)²⁴:

$$SF = \sum_{n=4}^6 \frac{\sum_{x=0}^n ((\omega_{calc}^{n,x} - \omega_{exp}^{n,x}) / \sigma_{integ})^2}{n + 1} \quad (5)$$

The errors of the mass peak integration are calculated using the RMSD (root-mean-square deviation) of the base line noise³⁵. Specifically:

$$\sigma_{integ}^2 = \sigma_i^2 * N \quad (6)$$

where the σ_{integ} is the standard deviation of the integration of peak intensity, σ_i is the standard deviation of the baseline noise, N is number of points in the region of integration.

Minor points:

1) In the supporting info characterizations are reported only for two samples, namely figures 2F and 5E. I presume that all the other samples analyzed (Figure 3, Figure 4, Figure 5A) are the very same of the original paper. In the case this is the case, this should be stated in the experimental section.

We have added in the Figure S2 for nanoparticles used in Figure 5A in the supporting information. The rest of the samples (13) used in this work are all PET-DDT silver nanoparticles with the same synthetic procedure and very similar size as characterized in Figure S2. We have clarified this point:

The nanoparticles were synthesized using the same procedure as described above therefore they have similar core sizes and only varies in the ratio of the two ligands.

2) In the nearest neighbour plots of figure 4, report the bars for both the ligands, as in those of figures 2 and 5.

We have added the nearest neighbour distribution plots for the other ligand in the Figure 4 as suggested by the referee.

3) The statement of “thin stripe-like domains on the NPs surfaces” in the case of the nanoparticles reported in figure 5F is overstretched (only two stripes 4 ligand long are visible) and should be removed. Indeed, the nearest neighbour plot (Figure 5G, letter G is missing) indicates that most (80% at least) of the blue ligands form just monomers or dimers.

We agree with the referee and have removed the statement of ‘stripe-like’. The following has been added:

One can see from the model that DDT ligand forms some dimers and trimers on the NPs surfaces, as also indicated by the nearest neighbor profile.

4) MALDI spectra for nanoparticles analyzed in figures 3, 4 and 5a should be reported in the ESI and it should be indicated which fragments were included in the analysis.

We have added all the MALDI spectra in the Figure S9 and specified the fragment peaks used for analysis, Table S1. In fact, we have uploaded all the raw data files (.txt) of the MALDI data to an open access data repository (figshare.com).

5) Distribution plots for the TEM pictures reported in the ESI should be reported (it looks unlikely that size dispersions of the particles in the two micrographs may be that similar, i.e. about 10% in both the cases, as in the second micrograph several very small particles are visible)

We have added the histogram of the size distribution from TEM analysis in Figure S2.

6) Fragment masses should be reported.

We have added Table S1 in the supporting information listing all the expected and experimental fragment mass regions used for integration. As also shown an example in Figure S4, the isotope pattern agrees very well with 50 ppm mass accuracy.

Reviewer #2 (Remarks to the Author):

The present manuscript by Stellacci and colleagues describes a methodology for using MALDI-TOFMS analyses of AuNP ligand distributions in conjunction with computer simulations to characterize the distribution of mixed-ligands on nanoparticle surfaces. The present work represents an extension of the series of works disseminated by Harkness et al. some years ago (Refs 26, 29, 31) along similar lines, but with several distinguishing differences. The present work is noteworthy in that it is demonstrated to also predict chemical shifts and correlations with SANS models are also presented. It also accounts for additional ligand species in the MS spectra, which may or may not improve the accuracy of the methodology.

We thank the referee for the overall positive overview of our work.

The addition of noise in the spectra and the impact on the conclusions drawn should be discussed. Along these lines, the present work makes no attempt to clean-up the “chemical noise” in the MALDI-TOFMS spectra, which was performed in the Harness series using ion mobility as an ion pre-separation prior to TOFMS analyses. The Stellacci manuscript could be improved by inclusion of a paragraph comparing the previous works and strategies to those presently presented. This should definitely be included, but does not diminish enthusiasm for the present manuscript.

We agree with the referee that IM-MS could help improve the signal-noise ratio of the spectra. On the other hand, we demonstrate that MS spectra with reasonable quality could also be obtained by basic MALDI-TOF setup. The same instrument was used in the work of McLean and co-workers. As MALDI is more accessible than IM-MS (indeed the latter is not available to us at EPFL), we think that it helps make this method a more routine approach.

Furthermore, as written in reply to referee #1:

“We now have improved the fitting algorithm taking into consideration the errors in the mass peak integration using an approach that is adopted by the Monte Carlo fitting of SANS spectra (Biophysical journal 76, no. 6 (1999): 2879-2886). In the Monte Carlo calculation, instead of simply using SSR as the scoring function, we now use normalized the SSR with the errors of each peak in the spectra. The new scoring function (SF) for experimental data now becomes:

$$SF = \sum_{n=4}^6 \frac{\sum_{x=0}^n ((\omega_{calc}^{n,x} - \omega_{exp}^{n,x}) / \sigma_{integ})^2}{n + 1}$$

The errors of the mass peak integration are calculated using the RMSD (root-mean-square deviation) of the base line noise (Methods Mol Biol. 2010, 673, 211–222). Specifically:

$$\sigma_{integ}^2 = \sigma_i^2 * N$$

where the σ_{integ} is the standard deviation of the integration of peak intensity, σ_i is the standard deviation of the baseline noise, N is number of points in the region of integration.

The models and nearest neighbour distribution profiles in all the figures are now updated by the results from the new fitting algorithm. The features of the ligand shell morphologies as well as the nearest neighbour distributions obtained are very similar compared to the previous version.”

We have changed/added the following content in the manuscript:

The main difference of fitting of experimental data compared to idealized models is the presence of errors in the measured of peak intensity and integration. Therefore, in the Monte Carlo calculation, instead of simply using SSR as the scoring function, the normalized SSR value by the errors of each peak in the spectra should be used as the new scoring function (SF)²⁴:

$$SF = \sum_{n=4}^6 \frac{\sum_{x=0}^n ((\omega_{calc}^{n,x} - \omega_{exp}^{n,x}) / \sigma_{integ})^2}{n + 1} \quad (5)$$

The errors of the mass peak integration are calculated using the RMSD (root-mean-square deviation) of the base line noise³⁵. Specifically:

$$\sigma_{integ}^2 = \sigma_i^2 * N \quad (6)$$

where the σ_{integ} is the standard deviation of the integration of peak intensity, σ_i is the standard deviation of the baseline noise, N is number of points in the region of integration.

The extension of these strategies through Monte Carlo simulations for convergence to obtain a more objective characterization and merging these analyses with correspondence to NMR chemical shift prediction and SANS models are well conducted and provide a more universal approach to the general characterization of Janus or phase-segregated nanoparticle systems. This should be of keen interest to the readership of Nature Communications.

We thank the referee for the positive outlook of the significance of our work.

Reviewer #3 (Remarks to the Author):

The current manuscript represents a notable contribution to a very active field of monolayers self-assembled on nanoparticles, a field with numerous experimental and theoretical studies including some that are conflicting. The main idea of the manuscript is to rapidly determine surface arrangement/morphology of binary ligand mixtures through reverse Monte Carlo fitting of MALDI-TOF data. Using a simple geometric descriptor (essentially, nearest neighbor distribution of ligands), the authors successfully reconstruct surface morphologies that explain the experimental MALDI data and provide quantitative indicators of specific morphologies (e.g., patchy vs Janus). Additionally, the method is validated through comparison to SANS data and is shown to be able to derive empirical NMR relationships for mixed-ligand NPs. Leveraging MALDI-TOF combined with theoretical modeling for morphology determination is not novel in the field, but this manuscript presents a much simpler and more general approach to surface reconstruction and can be recommended to publication in Nature Communications.

We thank the referee for his/her positive evaluation of the importance of our work.

However, before it is published several omissions should be fixed.

Overall, plots of nearest neighbor distribution histograms should be revised or clarified. While I understand that the y-axis represents fractions of ligands, nowhere in the text or the plots themselves (e.g., y-label) is it mentioned. Similarly, I find the x-label (“Number”) very vague. It is explained in the caption but I suggest using more scientific notation in the plots (e.g., $N_{nearest}$).

We have modified the label of axis and explained their meaning according to the referee's suggestion to make the plots more understandable.

Sticking to the histograms, can the authors comment on/show uncertainties for the data?

Thanks for the question raised by the referee. As written in reply to referee #1:

“ We now have improved the fitting algorithm taking into consideration the errors in the mass peak integration using an approach that is adopted by the Monte Carlo fitting of SANS spectra (Biophysical journal 76, no. 6 (1999): 2879-2886). In the Monte Carlo calculation, instead of simply using SSR as the scoring function, we now use normalized the SSR with the errors of each peak in the spectra. The new scoring function (SF) for experimental data now becomes:

$$SF = \sum_{n=4}^6 \frac{\sum_{x=0}^n ((\omega_{calc}^{n,x} - \omega_{exp}^{n,x}) / \sigma_{integ})^2}{n + 1}$$

The errors of the mass peak integration are calculated using the RMSD (root-mean-square deviation) of the base line noise (Methods Mol Biol. 2010, 673, 211–222). Specifically:

$$\sigma_{integ}^2 = \sigma_i^2 * N$$

where the σ_{integ} is the standard deviation of the integration of peak intensity, σ_i is the standard deviation of the baseline noise, N is number of points in the region of integration.

The models and nearest neighbour distribution profiles in all the figures are now updated by the results from the new fitting algorithm. The features of the ligand shell morphologies as well as the nearest neighbour distributions obtained are very similar compared to the previous version. ”

We have changed/added the following content in the manuscript:

The main difference of fitting of experimental data compared to idealized models is the presence of errors in the measured of peak intensity and integration. Therefore, in the Monte Carlo calculation, instead of simply using SSR as the scoring function, the normalized SSR value by the errors of each peak in the spectra should be used as the new scoring function (SF)²⁴:

$$SF = \sum_{n=4}^6 \frac{\sum_{x=0}^n ((\omega_{calc}^{n,x} - \omega_{exp}^{n,x}) / \sigma_{integ})^2}{n + 1} \quad (5)$$

The errors of the mass peak integration are calculated using the RMSD (root-mean-square deviation) of the base line noise³⁵. Specifically:

$$\sigma_{integ}^2 = \sigma_i^2 * N \quad (6)$$

where the σ_{integ} is the standard deviation of the integration of peak intensity, σ_i is the standard deviation of the baseline noise, N is number of points in the region of integration.

Considering significant error bars shown in Fig. 1, it will be interesting to see how sensitive the nearest neighbor descriptor is to small variations in the surface arrangement. It should be easy to generate the error bars considering how fast MC fitting is.

Thanks to the referee's suggestion, we have repeated the Monte Carlo calculations multiple times (>5) for all the nanoparticles studied in this work. The generated models could be regarded as representations in the space of possible structures corresponding to each MALDI spectra. As shown in the updated Figure 2 and Figure 3, the standard deviations in nearest neighbour distributions are calculated to quantify the variations among all the possible models below the SSR threshold.

In order to clarify this issue, we have added the following:

Furthermore, as one could notice from Figure 1A-D that the models from Monte Carlo calculations do not always capture all the structural details in complex morphologies. This is unfortunately due to the intrinsic ambiguity in the structural information that the MALDI data contains. Different models could give very similar experimental footprint and thus the solution from Monte Carlo fitting is not unique. In order to address this limitation, the fitting program was repeated 5 times for each morphology. Each generated model can be regarded as a possible solution in the space of all possible structures corresponding to a given MALDI spectra. It is important to notice that the true physical cause for the MALDI spectra is the nearest neighbor distribution

and this is the true result of our method. Hence for all of the particles studied, we have generated standard deviations in nearest neighbor distributions. As shown in Figure 1A-D, the standard deviations for all the calculations are small indicating that all the models from repeated calculations are very similar to each other. These standard deviations are the best way to estimate the resolution of our model. Ligand shell morphologies that produce nearest neighbour distribution that fall within these standard deviations will not be distinguishable with this approach. We should point out that our empirical observation is that SSR values that are within 10^{-4} to each other lead to indistinguishable nearest neighbor distributions.

Similarly, it is unclear what the error bars in Fig. 1 are intended to represent. Do they come from multiple Monte Carlo runs? Are they block averages? This needs to be explained.

The error bars in Figure 1 indeed come from multiple Monte Carlo runs and are the standard deviations of SSR values after certain iteration steps.

We have explained the meaning of the error bars in the manuscript:

The Monte Carlo program was run multiple times (> 10) starting always from random configurations, i.e. binomial distribution and gradually converges to the input pattern. The error bars in Figure 1A are the standard deviations of SSR values after certain iteration steps.

Comparing histograms and surface arrangement of the ligands in Figs. 1D and 1G, authors state that nearest neighbor distributions are very similar, which I agree with. However, I cannot fully agree that ligand morphologies are the same. The authors themselves say that particle G has considerable patches and stripes while calculated particle D exhibits only thin stripes. To further understand the suitability of the nearest neighbor descriptor and its possible limitations in comparing different predictions, the authors should look into more complex descriptors like nearest neighbor distribution in the first two neighboring shells instead of one (18 neighbors vs 6) like it was done in Fetisov and Siepmann, J. Phys. Chem. B, p. 1972, 2016. Although it will not change how the Monte Carlo fits are performed since only the first neighboring shell matters for MALDI-TOF spectra fitting, it can help to further disseminate the results or determine limitation of the method.

We thank the referee for the great suggestion. We have calculated the nearest neighbour distribution of the first two neighbouring shells (18 neighbours) referencing the literature that the referee mentioned. The comparison of the models in Figure 1D and 1G is shown in Figure S7. It could be seen that indeed although the first nearest neighbour (6) distribution of the two models are similar, when 18 neighbours are taken into consideration, the differences between the two models become clearer. While the idealized stripe-like nanoparticle shows a more centralized distribution featuring the high fraction of 7-10 same nearest neighbours, the distribution profile for PET-DDT nanoparticle is broader.

We have modified/added the above discussion in the manuscript:

As shown in Figure 2G, the resulted model shows a complicated organization of the two ligands, featuring both small patches as well as stripe-like domains. As a result, the nearest neighbor distribution of this nanoparticle is close to that of idealized stripe-like LSM in Figure 2D compared to other morphologies. This is due to the fact that no large patchy structures of the ligands are formed. To further understand the suitability of the nearest neighbor descriptor and its possible limitations in comparing different predictions, a more complex descriptor, i.e. nearest neighbor distribution in the first two neighboring shells (18 neighbors) of the two models are computed and compared. The same type of analysis was reported previously³⁶. As shown in Figure S7, in the 18 nearest neighbor distribution profile, the differences between the two structures become clearer. While the idealized stripe-like nanoparticle shows a more centralized distribution featuring the high fraction of 7-10 same nearest neighbors, the distribution profile for PET-DDT nanoparticle is broader.

This leads me to the main omission in the presentation of the idealized Monte Carlo fitting results. It is well known that a main limitation of reverse Monte Carlo methods is the possibility of different models giving similar experimental footprint. While authors can successfully arrive to specific morphologies starting for random arrangements, what happens if one wants to fit to a random morphology? Is the

developed MC method robust enough to reproduce the binominal distribution or will it possibly lead to some higher symmetry?

We completely agree with the referee about the limitations of the Reverse Monte Carlo methods. We have added a sentence in the manuscript to stress again that the fitting needs to be run for multiple times to compare the retrieved models in order to better understand the space of structures that the experimental data correspond.

We have also performed the fitting to a random morphology. As shown in Figure S1, the MC method could reproduce the random bead arrangement as well as the close to binomial distribution.

The following text has been added to clarify this point:

Meanwhile, the fitting procedure is based on random switch of bead assignments and does not lead to higher symmetry. As an example, the program is used to fit the spectra coming from a random geometry. As shown in Figure S1. The Monte Carlo calculation could reproduce the random bead arrangement as well as the close to binomial distribution of nearest neighbor.

On p.6 at the end of the second paragraph, I find the discussion about different SSR values a bit confusing and I suggest further clarification for readers. In the literature, SSR values are almost exclusively computed with respect to binominal distribution of ligands. In this manuscript, the authors discuss SSR with respect to idealized Janus topology and state that value of 10^{-3} would correspond to patchy NPs. However, this is only true if one uses Janus arrangement as the reference, but not binominal. This should be clarified since the SSR scale is flipped with respect to published trends.

We thank the referee for pointing out the confusion in our description. Indeed, in that paragraph, we are discussing about SSR against Janus arrangement. We have stressed and clarified it in the manuscript. The text has been added/modified:

Note that compared to previous literatures in which SSR values are calculated based on binomial distribution, here the SSR value represents the statistical distance between a model and a perfect Janus arrangement.

When discussing quality of fits for different morphologies on p.7, the authors say that the SSR value for all fits was below 2×10^{-5} . Can authors explain why the fits are suddenly much better here, since throughout the whole text all SSR values were on the order of 10^{-4} or more, especially considering the accuracy (error bars)?

The reason for the increase of SSR in the experimental data compared to idealized morphology indeed comes from the errors in the experimental measurements. In the idealized morphology, the mass peak intensities of the outcome model overlap almost perfectly with the input spectra as there was no errors and uncertainties.

We have added the following to clarify this:

The slight increase of final SSR values compared to the idealized geometry is due to the presence of errors in the experimental data. In the idealized morphology, the mass peak intensities of the outcome model overlap almost perfectly with the input spectra as there was no errors and uncertainties.

Overall, the section "Test on various samples" is contextually sparse. First, there is no mention of the experimental sizes and ligand ratios for the presented Ag NPs, which limits reproducibility. Secondly, the authors note that these NPs have been extensively studied in the literature but they do not discuss their findings in the context of the previous results. Are they consistent? Similarly, the paragraph about Au NPs does not bring anything new to the manuscript without comparing to previous findings. By this point, it is already clear that the approach developed by the authors can successfully reproduce experimental spectra and the mention of Au NPs detracts from the primary story. What would be more interesting is to discuss if, when applied to Au NPs, the new method also reveals something new or at least confirms previous findings/theories. Otherwise, I suggest to move this part to the Supporting Information.

First, we have added the missing information including the size and distribution (together with TEM image), ligand ratios for the AgNPs in the text. We agree with the referee and have moved this last section to the Supporting Information. We added in the previous version just to show that the method is not restricted to silver core.

Also, the nearest neighbor distributions and the image of the Au NP suggest that the arrangement is more close to random distribution of monomers/"dimers" of DDT in MUA instead of stripe-like domains.

We agree with the referee and have removed the statement of 'stripe-like'. The following has been added:

One can see from the model that DDT ligand forms some dimers and trimers on the NPs surfaces, as also indicated by the nearest neighbor profile.

And some relatively minor but still important remarks:

1) On p.2, the authors write "ab initio calculations, i.e. Monte Carlo based methods". I believe there is a misuse of the phrase "ab initio" here that can lead to confusion.

We have removed the "ab initio" in the sentence to avoid confusion.

2) On p.3, the authors discuss $M_k L_x L_n - x$ fragments and say that constant n leads to constant x . In the next sentence, they say that k varies from 0 to n to give a binominal distribution. Should not x and k be exchanged in this context?

We are sorry for the mistake. The x and k should be exchanged. Now the sentence is fixed.

3) On p.9, "to quantitatively characterization" should read "to quantitatively characterize".

We have changed it.

4) Caption for Fig.5 is missing part (E) and instead has additional part (G).

We have added the missing parts in Figure 5.

5) On p.10, the first two sentences in the second paragraph have to be combined.

We have combined it. Now the sentence reads:

Assuming that the final chemical shift of the ligand on the nanoparticle is the sum over all the possible situations of its nearest neighbor distribution (n_i), one could get:

REVIEWERS' COMMENTS:

Reviewer #1 (Remarks to the Author):

The first version of the manuscript obtained quite unanimous positive comments by all the referees, confirming the potential high impact of the research described. In this revised version, the authors have fully addressed all the points raised. For these reasons, I believe that the paper deserve publication on Nature Communications without further modifications.

Reviewer #3 (Remarks to the Author):

The referee points have been satisfactorily addressed and the manuscript can be published.

However, as the last step, authors should double check their Figure numbering throughout the text and the SI. Some references do not correspond to the correct figures.

Response to reviewer's comments:

Reviewer #1 (Remarks to the Author):

The first version of the manuscript obtained quite unanimous positive comments by all the referees, confirming the potential high impact of the research described. In this revised version, the authors have fully addressed all the points raised. For these reasons, I believe that the paper deserve publication on Nature Communications without further modifications.

We thank the referee for the positive evaluation and constructive comments.

Reviewer #3 (Remarks to the Author):

The referee points have been satisfactorily addressed and the manuscript can be published.

We thank the referee for the assessment.

However, as the last step, authors should double check their Figure numbering throughout the text and the SI. Some references do not correspond to the correct figures.

We have carefully checked all the numbering of figures in the revised manuscript and SI.